# Optimisation Models for Inventory Management with Limited Number of Stock Items

Julian Vasilev [1],* and Tanka Milkova [2],*

1 Department of Informatics, University of Economics Varna, 9002 Varna, Bulgaria
2 Department of Statistics and Applied Mathematics, University of Economics Varna, 9002 Varna, Bulgaria
* Correspondence: vasilev@ue-varna.bg (J.V.); tankamilkova@ue-varna.bg (T.M.)

**Abstract:** *Background*: Stocks of raw materials and finished products are found in all units of logistics systems and require significant financial means of management. For this reason, scientifically justified approaches to stock management and cost minimisation must be explored. Despite the existence of many such approaches in literature and practice, each case has its own specificities and specificities to which stock management models should be adapted. In this article, the aim of the authors is to propose an approach to determine optimal supply sizes from different types of stocks (more than one is known in the literature as multi-nomenclature) that minimises only the cost of inventory management. The cost of inventory is not included. *Methods*: The article used the methods of mathematical optimisation, the method of least squares, and regression analysis. The scope of the models in the article is inventory management, with a limited number of stock keeping units. Time series data for the delivered quantities and time series data for the costs of stock management are used. Both time series use the same time period. *Results*: The constructed specific nonlinear mathematical models for optimising the total cost of stock management are approbated based on sample data and the results obtained are analysed. *Conclusions*: The created mathematical models and methods for optimising the total cost of stock management may be used by logistics managers to minimise the total costs of inventory management.

**Keywords:** optimisation; inventory management; logistic system



## 1. Introduction

The continuous implementation of a set of complex processes and activities is a prerequisite for ensuring the effective functioning of any business organisation in the modern economy. Optimal inventory management is essential among this set of activities. Given that stocks represent the static state of material flow, they occur in each logistics system. The dynamic environment in which modern organisations must work determines the continuous theoretical and practical interest in the problems related to stock management. According to some studies, the stocks present everywhere in the logistics systems command up to 80% of the working resources of the organisations, and the costs of maintaining and managing them reach up to 40% of the logistics costs of the companies, in a number of cases [1,2].

According to one of the required definitions of inventories, they are located in different stages of production and circulation of production and technical production, consumer and other goods awaiting entry into the production process or consumption, occurring in all phases of the business process—in the supply (raw materials, complete products), in the production process (incomplete production, instruments, semi-finished products, etc.), and in the market (finished products, spare parts, accompanying goods for the workshops) [3,4].

The availability of stock is associated with both positive effects on the organisation's activities and various negative financial implications.

Some of the main advantages of stock formation consists of the following: ensuring business continuity (especially in production processes), the possibility of immediate customer service, the availability of commercial discounts for the purchase of large quantities of cargo, simplification of the production management process, and protection from price increases from unfair suppliers, etc.

The most significant negative effects of the availability of stock in all stages of production and circulation are as follows: freezing of a significant financial resource, commitment of human resources, and additional management costs, etc.

Given that stock in the logistics system has both positive and negative aspects, there is a need to use scientifically sound approaches to inventory management to achieve better economic results [5,6]. In specialised literature and business practice, numerous models [7–9] and methods [10–12] for managing inventories are known, depending on the type of stock, the place of its formation, and the nature of its consumption, specific methods are recommended for use. Nevertheless, there are still a number of specific features related to inventory management that are not covered by common models and methods, but need specific, modified models to more accurately describe them.

The well-known Wilson's model for inventory management [13] has parameters (covariates) describing different costs components for inventory management. These parameters are difficult for defining and data collection. Their values often differ from real business values. A novelty in this paper is the offered approach for constructing the total cost's function based on real (business) data. These data are aggregated for past periods (historical data). The proposed approach (by the authors of this article) gives the possibility of using the total cost's function to describe the real situation more precisely. The drawbacks of the Wilson's model for inventory management are solved by the proposed approach.

The aim of the authors in this article is to propose an approach to determine optimal supply sizes from different types of stock (more than one is known in the literature as multi-nomenclature) that minimises only the cost of inventory management. The cost of inventory is not included.

## 2. Literature Review

Teachers [14–16] who teach mathematics [17–19] and logistics in education [20–23] usually try to find mathematical models that can be adapted in practices, that may be validated with real business data or samples with data. The know-how of the adoption of such models gives students (and, respectively, future businessmen/businesswomen) the confidence to adapt new models which may help in reducing overall costs and bringing to the market products with competitive prices. These models need to have a proper software adaption [24–28]. Using programming languages [29–31], the source code may be written. Furthermore, web applications [32–35] may be created with a built-in business logic for inventory management. The proposed method for inventory management has to be compared to other similar methods to check its performance [36,37]. Educational services need further digitalization [38,39]. In this context, the newly created approaches may be useful for students as well as for practitioners.

Many articles focus on "optimisation models" for "inventory management" [40]. Even though many findings are clear for researchers nowadays, the search for new and better models to optimise inventory management continues. Some of them are focused on a specific supply chain (such as pharmaceuticals) [41]. A great number of articles are focused on green models [42]. Other models focus on the interactions of deterministic and stochastic optimisation models in information sharing in supply chains [43]. Some models are single stock models [44], others are multi stock models [45]. Using regression function for inventory management is common practice [46–48]. The Wilson's model for inventory management is well-known and adapted by many researchers [49–53], even though some drawbacks and limitations still exist. In some cases, practitioners declare that they have difficulties in adapting theoretical models with many restrictions and limitations in practices.

The proposed models by the authors of this article are built with the assumption of a limited number of stock items. Companies which have a big nomenclature of stock items may make a sample of them and analyse it with the proposed models. They will need to have observations for real costs to create the historic dataset. This dataset must be created for each observed item and for fixed time periods, e.g., days, months, and quarters. One of the novelties of the proposed models is the use of real historical data for different cost components for constructing the cost's function for inventory management.

Section 1 is the introduction, it describes the problem and the real gap in inventory management. Section 2 deals with the literature review and the current state of the research problem. Section 3 describes the proposed models, methods, and the research design. Results and discussion are given in Section 4. Conclusions and an outline for further research are given in Section 5.

## 3. Methods and Research Design

In general, two main types of consumption of stock are known in theory and practice, namely:

(1) Regular (even)—the stock is used daily, weekly, monthly, etc. In the management of stocks that have this kind of need, various modifications of fundamentals in the theory of inventory management is needed, and Wilson's model [13,54] is recommended.
(2) Irregular (uneven)—inventory demand occurs at random times (e.g., spare parts). Then the demand forecast cannot be made with high levels of accuracy, and different models for managing stock are recommended in case of accidental demand [55,56].

In both types of inventory consumption, the following dependency is observed. The total cost of managing inventory depends on the size of a shipment and includes some basic components. One group of costs relates to stock storage and is proportional to the size of the supply of stock, i.e., higher storage costs are formed at a larger stock size. Other types of cost—which are either proportional or inversely proportional to the size of stocks—shall also be considered. In any event, the total costs formed may be modelled by a function which has a parabola type and has a minimum.

This article suggests an approach to determine optimal stock sizes of different types, subject to the following restrictions:

(1) Deliveries are made at certain times at regular intervals—for example, once a week, monthly, etc.
(2) The consumption of stocks is even.
(3) High storage costs and, at the same time, high costs associated with stock shortages are observed.
(4) In the case of a shortage of stock, needs cannot be met when the next supply is received.

### 3.1. Staging of the Optimisation Model in the Presence of a Single Stock

In the case of a stock in volume of x units of a certain product, it is possible to generate costs (just for stock management, not including the costs of the inventory itself) in the amount of c currency units. These costs can be accumulated for two main reasons: storage costs and costs from under-stock.

Let the volumes $x_i$ and therefore costs for past deliveries ($c_i$, i = 1, 2, ... , m) already made known, which are carried out at m regular/fixed intervals (e.g., monthly).

The costs are a one-dimensional array $\{c_1, c_2, \ldots, c_m\}$. The volumes of deliveries are also a one-dimensional array $\{x_1, x_2, \ldots, x_m\}$. A possible algorithm for finding the optimal solution is the following: The minimal element in the array is found $\{c_1, c_2, \ldots, c_m\}$. Its position in the array is stored in the variable "pos". The optimal volume for stock delivery is $x_{pos}$.

The novelty in the proposed model in Section 3.1 (focused on finding the optimal solution just for one stock) is proposing a fast algorithm for finding the volume of stock delivery with minimal inventory costs (not including the cost of the inventory itself). The model in Section 3.1. has no additional limitations for the stock volumes.

From a practical point of view, it is normal for small and large values of x that the cost value c is high. In the first case, due to high costs of shortage and the need for additional supplies, and in the second, due to storage costs, etc.

### 3.2. Staging of the Optimisation Model in the Presence of More than One Type of Stock and Where No Restrictions Are Imposed

Now, we will consider the situation with multi-product stocks. Let the $n$ stock type be present: $A_1, A_2, \ldots, A_n$, the volumes for each we will indicate with $x^j$, j = 1, 2, . . . , $n$ and the costs of each with $c^j$, j = 1, 2, . . . , $n$.

Data for the supplied volumes are time series data. Data for the costs of stock management are also time series data. Both time series use the same time period. Filling in both datasets with this important issue guarantees the correctness of the model. Creating a sample dataset from real business data may be performed in 3 steps. Firstly, choosing stock items with fixed-period delivery (e.g., monthly). Secondly, if the list is too big for data collection and data analysis a smaller sample may be chosen. Thirdly, time series data for costs management for each stock unit must be collected (e.g., monthly). Since companies have confidential data, sample datasets are created by the authors to illustrate the numeric examples.

The quantities of $x_i{}^j$, i = 1, 2, . . . , m; j = 1, 2, . . . , $n$ stocks of each product type and the corresponding $c_i{}^j$, i = 1, 2, . . . , m; j = 1, 2, . . . , $n$ costs for each period are known for m periods, depending on the volume of stock of each product type for each of the periods.

Deliveries are made at regular intervals and the data is summarized in the following tables (Tables 1 and 2).

**Table 1.** Quantity of stocks.

| $\begin{matrix}j\\i\end{matrix}$ | $x^1$ | $x^2$ | ... | $x^j$ | ... | $x^n$ |
|---|---|---|---|---|---|---|
| 1 | $x_1{}^1$ | $x_1{}^2$ | ... | $x_1{}^j$ | ... | $x_1{}^n$ |
| 2 | $x_2{}^1$ | $x_2{}^2$ | ... | $x_2{}^j$ | ... | $x_2{}^n$ |
| ... | ... | ... | ... | ... | ... | ... |
| i | $x_i{}^1$ | $x_i{}^2$ | ... | $x_i{}^j$ | ... | $x_i{}^n$ |
| ... | ... | ... | ... | ... | ... | ... |
| m | $x_m{}^1$ | $x_m{}^2$ | ... | $x_m{}^j$ | ... | $x_m{}^n$ |

**Table 2.** Total cost.

| $\begin{matrix}j\\i\end{matrix}$ | $c^1$ | $c^2$ | ... | $c^j$ | ... | $c^n$ |
|---|---|---|---|---|---|---|
| 1 | $c_1{}^1$ | $c_1{}^2$ | ... | $c_1{}^j$ | ... | $c_1{}^n$ |
| 2 | $c_2{}^1$ | $c_2{}^2$ | ... | $c_2{}^j$ | ... | $c_2{}^n$ |
| ... | ... | ... | ... | ... | ... | ... |
| i | $c_i{}^1$ | $c_i{}^2$ | ... | $c_i{}^j$ | ... | $c_i{}^n$ |
| ... | ... | ... | ... | ... | ... | ... |
| m | $c_m{}^1$ | $c_m{}^2$ | ... | $c_m{}^j$ | ... | $c_m{}^n$ |

Here, we assume that there are no dependencies and limitations between the volumes of different types of stocks. In this condition, the optimal stock of j type $x^j{}_{i0(j)}$, j = 1, 2, . . . , $n$ shall be determined by: $i_0(j)$ is such that $c^j{}_{i0(j)} = \min\{c_1{}^j, c_2{}^j, \ldots, c_m{}^j\}$ for each j = 1, 2, . . . , $n$.

Here, as in 2.1, on the basis of practice, we assume that $x^j{}_{i0(j)}$ is one of the intermediate values of $x_i{}^j$, i = 1, 2, . . . , m for each j = 1, 2, . . . , $n$, since in the case of small and large $x_i{}^j$, i = 1, 2, . . . , m the costs are higher.

This means that in Table 2 in each column we determine the smallest number and find the number that stands in the same position but in Table 1. This is how we determine all the optimal volumes $x^j{}_{i0(j)}$ for each j = 1, 2, . . . , $n$.

Let us look at an example of sample data.

Four products were delivered to storage at the beginning of each month during the previous year, the quantities of which are given in Table 3.

**Table 3.** Quantity of supplied stocks of four products for the previous 12 months (in kg).

| Product | $A^1$ | $A^2$ | $A^3$ | $A^4$ |
|---|---|---|---|---|
| $x^i$ Month | $x^1$ | $x^2$ | $x^3$ | $x^4$ |
| 1 | 200 | 300 | 180 | 250 |
| 2 | 120 | 220 | 210 | 230 |
| 3 | 180 | 250 | 160 | 235 |
| 4 | 130 | 240 | 140 | 260 |
| 5 | 240 | 190 | 110 | 320 |
| 6 | 260 | 210 | 150 | 180 |
| 7 | 210 | 280 | 170 | 200 |
| 8 | 150 | 310 | 200 | 210 |
| 9 | 110 | 245 | 215 | 170 |
| 10 | 165 | 315 | 165 | 150 |
| 11 | 170 | 230 | 130 | 205 |
| 12 | 160 | 180 | 120 | 215 |

The corresponding costs, depending on the level of the stock, are given in Table 4.

**Table 4.** Costs (in BGN) for inventory management for the previous 12months.

| Costs Month | $c^1$ | $c^2$ | $c^3$ | $c^4$ |
|---|---|---|---|---|
| 1 | 130 | 180 | 110 | 160 |
| 2 | 140 | 110 | 125 | 145 |
| 3 | 100 | 130 | 95 | 150 |
| 4 | 125 | 120 | 105 | 170 |
| 5 | 170 | 150 | 130 | 210 |
| 6 | 180 | 115 | 90 | 155 |
| 7 | 135 | 160 | 100 | 130 |
| 8 | 130 | 170 | 120 | 120 |
| 9 | 120 | 135 | 135 | 165 |
| 10 | 110 | 165 | 98 | 180 |
| 11 | 105 | 117 | 108 | 125 |
| 12 | 115 | 155 | 118 | 122 |

In Table 4, the minimum column elements are $c_3^1 = 100$, $c_2^2 = 110$, $c_6^3 = 90$ and $c_8^4 = 120$, respectively. This means that if at the beginning of each month a stock of the products of the first 180 kg, the second 220 kg, the third 150 kg, and the fourth 210 kg is carried out, then it can be expected that the total costs will be minimal and will be about $c_3^1 + c_2^2 + c_6^3 + c_8^4 = 100 + 110 + 90 + 120 = 420$ BGN per month.

Again, we would like to stress that this approach is applicable when there is no dependence between the volumes of stocks of individual products, i.e., no restrictions have been imposed on variables $x_i^j$, i = 1, 2, . . . , m; j = 1, 2, . . . , *n*, except for non-negative conditions.

The novelty of the proposed model in Section 3.2 is the possibility of constructing the cost's function with several stocks. As a result, the model finds the quantities of each stock where the inventory costs (not including the cost for the inventory itself) are minimal. The model in Section 3.2 has no additional limitations for the stock volumes.



*3.3. Staging of the Optimisation Model in the Presence of More than One Type of Stock and the Presence of Restrictions*

Often in practice there is a need to impose restrictions. Reasons for this may be the capacity of the warehouse, for example, or the imposition of requirements for minimum (and/or maximum) volume of stock of a given type, etc.

Let us now assume that, in the example we set out in 2.2, there is an additional restrictive condition, namely the capacity of the warehouse is 700 kg. Then the optimal solution we have received is not acceptable, since $x_3{}^1 + x_2{}^2 + x_6{}^3 + x_8{}^4 = 180 + 220 + 150 + 210 = 760$ kg, which exceeds the capacity of the warehouse and it is not possible to store these quantities of these stocks of the four products on a monthly basis. Of course, we can search Table 3 for other values close to the optimum values, so that their amount does not exceed 700 kg. However, it is difficult to apply this method to a large number of historical data and a large number of products.

Therefore, here we offer a method for solving a multi-product task for managing stocks with restrictions.

First, if, indeed, at small and large values per supply, the costs are greater than for the intermediate values of the amount of supply of a given type of stock, then this means that the dependence of costs c on the quantity of stock x is square and the method of least squares can be used to obtain the analytical type of this dependency [57,58].

Secondly, for each product j, using the data from column $A_j$, $j = 1, 2, \ldots, n$ in Tables 1 and 2 by square regression, we obtain the dependency:

$$c^j (x^j) = a^j + b^j x^j + d^j (x^j)^2, j = 1, 2, \ldots, n$$

Thirdly, referring to the staging of the task and the theoretical productions of mathematical optimisation [59,60], namely minimising the total costs, we construct the following mathematical model:

$$\min: Z (x^1, x^2, \ldots, x^n) = a^1 + b^1 x^1 + d^1(x^1)^2 + a^2 + b^2 x^2 + d^2(x^2)^2 + \ldots$$
$$+ a^n + b^n x^n + d^n(x^n)^2 \tag{1}$$

under restrictive conditions:

$$x^1 + x^2 + \ldots + x^n \leq Q \tag{2}$$

$$x^j \geq 0, j = 1, 2, \ldots, n \tag{3}$$

where the capacity of the warehouse is Q.

We would like to make some clarifications. First, it is possible to include other restrictive conditions that reflect the real situation in inventory management. Such an additional restrictive condition may be

$$v_j \leq x_j \leq V_j, j = 1, 2, \ldots, n \tag{4}$$

if a requirement is required that delivered volumes from each stock must exceed a specified minimum level ($v_j$) and not exceed another specified level ($V_j$).

The following clarification relates to the fact that a restrictive condition (2) is in the form of inequality, which is more general than in the form of equality, because there may not be a requirement to fully fill the capacity of the warehouse.

We will present the application of the model thus constructed with the data from the example by imposing the additional capacity limit of the warehouse in the amount of 700 kg.

First, we will present the values of the dimensions of the deliveries of each type of stock in Table 3 (from each column) in ascending order and we will present the rearranged corresponding values of the total cost of inventory management (Table 4) corresponding to these dimensions of monthly deliveries. Data for stock level and total cost of all products are given in Table 5.

**Table 5.** Stock level and total cost of all products.

| $A_1$ | | $A_2$ | | $A_3$ | | $A_4$ | |
|---|---|---|---|---|---|---|---|
| $x^1$ | $c^1$ | $x^2$ | $c^2$ | $x^3$ | $c^3$ | $x^4$ | $c^4$ |
| 110 | 120 | 180 | 155 | 110 | 130 | 150 | 180 |
| 120 | 140 | 190 | 150 | 120 | 118 | 170 | 165 |
| 130 | 125 | 210 | 115 | 130 | 108 | 180 | 155 |
| 150 | 130 | 220 | 110 | 140 | 105 | 200 | 130 |
| 160 | 115 | 230 | 117 | 150 | 90 | 205 | 125 |
| 165 | 110 | 240 | 120 | 160 | 95 | 210 | 120 |
| 170 | 105 | 245 | 135 | 165 | 98 | 215 | 122 |
| 180 | 100 | 250 | 130 | 170 | 100 | 230 | 145 |
| 200 | 130 | 280 | 160 | 180 | 110 | 235 | 150 |
| 210 | 135 | 300 | 180 | 200 | 120 | 250 | 160 |
| 240 | 170 | 310 | 170 | 210 | 125 | 260 | 170 |
| 260 | 180 | 315 | 165 | 215 | 135 | 320 | 210 |

Based on this data, the following analytical models are obtained using MS Excel (Figures 1–4):

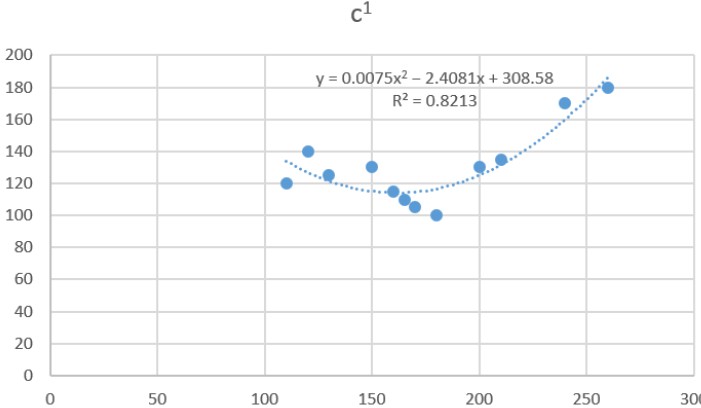

**Figure 1.** Analytical cost dependency on the size of the stock of the first type.

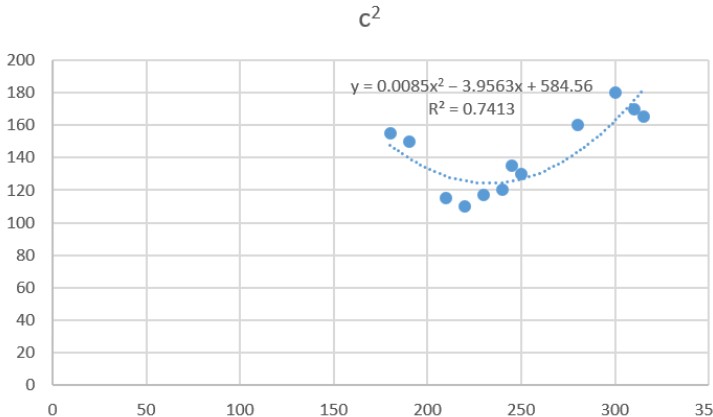

**Figure 2.** Analytical cost dependency on the size of the stock of the second type.

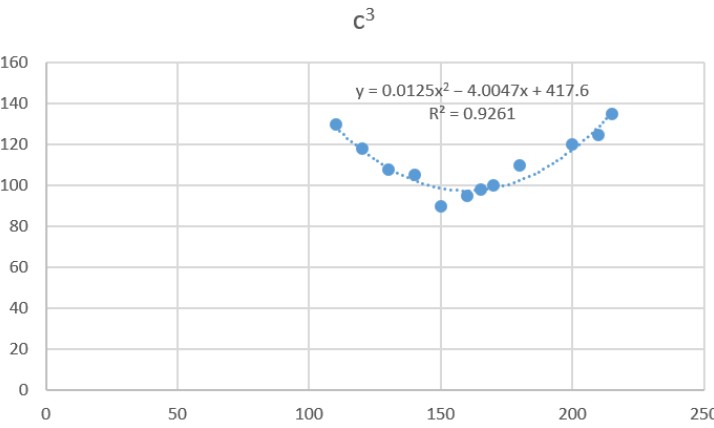

**Figure 3.** Analytical cost dependency on the size of the stock of the third type.

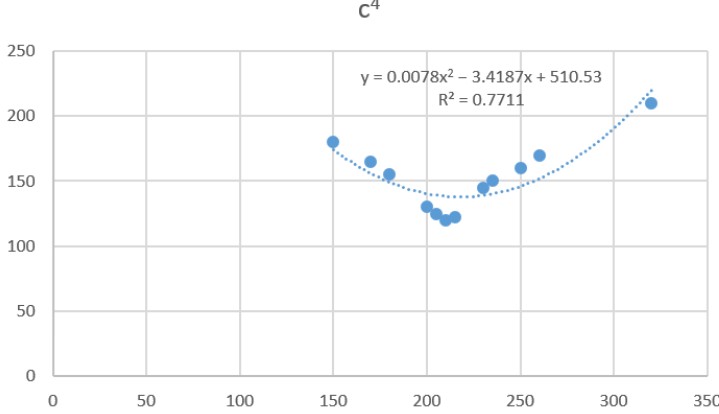

**Figure 4.** Analytical cost dependency on the size of the stock of the fourth type.

Thus, the functions describing analytically the dependence of costs on the quantity of stocks of each type shall accept the type:

$$c^1 = 0.0075\,(x^1)^2 - 2.4081\,x^1 + 308.58, \qquad\qquad R^2 = 0.8213,$$
$$c^2 = 0.0085\,(x^2)^2 - 3.9563\,x^2 + 584.56, \qquad\qquad R^2 = 0.7413,$$
$$c^3 = 0.0125\,(x^3)^2 - 4.0047\,x^3 + 417.6, \qquad\qquad R^2 = 0.9261,$$
$$c^4 = 0.0078\,(x^4)^2 - 3.4187\,x^4 + 510.53, \qquad\qquad R^2 = 0.7711.$$

Each of the four models has a sufficiently high value of the coefficient of detergent, and we can assume these models are reliable.

The model of the optimisation task (1)–(3) accepts the type:

$$\min: Z\,(x^1, x^2, x^3, x^4) = 0.0075\,(x^1)^2 - 2.4081\,x^1 + 308.58 +$$
$$0.0085\,(x^2)^2 - 3.9563\,x^2 + 584.56 +$$
$$0.0125\,(x^3)^2 - 4.0047\,x^3 + 417.6 +$$
$$0.0078\,(x^4)^2 - 3.4187\,x^4 + 510.53$$

under restrictive conditions:

$$x^1 + x^2 + x^3 + x^4 \le 700,$$
$$x^1 \ge 0,\ x^2 \ge 0,\ x^3 \ge 0,\ x^4 \ge 0$$

The novelty of the proposed model in Section 3.3 is the possibility of constructing the cost's function with several stocks. The model assumes that the total volumes of stock reserves of all items must not exceed the warehouse capacity. As a result, the model finds the quantities of each stock where the inventory costs (not including the cost for the

inventory itself) are minimal. The model in Section 3.3 has no additional limitations for the stock volumes.

## 4. Results and Discussion

The optimal solution to this optimization task is determined by using MS Excel and is as follows using the proposed model in Section 3.3:

$$x^1 = 139.46, x^2 = 214.12, x^3 = 147.54, x^4 = 198.88 \text{ and min: } Z = 483.74$$

Of course, the values thus obtained for the dimensions of supplies from stocks may be rounded per $x^1 = 139$ kg., $x^2 = 214$ kg., $x^3 = 148$ kg., $x^4 = 199$ kg., which is actually the optimal integer solution to the task. This means that supplies of 139 kg must be made from the first type of stock, supplies of 214 kg must be made from the second type of stock, the third type of stock must be supplied at a rate of 148 kg, and the fourth type of stock must be supplied at a rate of 199 kg. This value is naturally higher than the minimum of the function, provided that there are no limits on the number of supplies of each type of stock, it is the optimal solution in case of restrictions.

The application of the proposed models allows the calculation of optimal size for each stock unit; minimal total costs for inventory (storage and lack/out-of-stock). This approach is better than Wilson's model. The total cost's function is constructed based on real data for inventory costs (not including the costs of the inventory itself). The proposed model works with a limited number of stocks. However, if it gives the optimal result timely then the list with stocks (included in the model) may be extended. Since the proposed model uses historical business data for the costs of stock reserves and costs for out-of-stock goods, other practitioners may replicate the proposed models with their business data.

The restriction for minimal and maximal volume for each stock item is given in the model in Section 3.3 but it is not given in the numerical example. Another novelty of the model in Section 3.3 is that the model may be extended, tested, and validated with other limitations.

## 5. Conclusions

The models presented in this article are based on sample data, but in the presence of real data they can be useful tools for determining an optimal strategy for replenishing the stock of multiple products, subject to restrictions. However, it must be observed whether the actual situation meets the conditions and nature of the demand for the stocks, for which this model is constructed.

The novelties of the proposed models have several aspects. Firstly, they extend upon Wilson's model for inventory management. Secondly, the use of real historical data for different cost components for constructing the cost's function for inventory management, and that the models may be extended, tested and validated with other restrictions.

The created models have practical implications. Logistics managers may choose a sample from their big items/stocks/inventory list. They may create time series data for delivered volumes (by stocks and by months), and time series data for the costs of inventory management (by stocks and by months). They may apply the proposed models to minimise the total costs for stock management. If the optimization models work fast, the logistics managers may extend the sample with more items.

As a guideline for the development and extension of the options for the application of the model, the possibility of adding further restrictive conditions relating, for example, to the maximum permissible total value of products stored in the form of stocks may be indicated.

Further research may be focused on creating new models based on the proposed ones where the time series data for delivery volumes has different time periods than the time series for the costs of inventory management (not including the costs of the inventory itself).

**Author Contributions:** J.V. and T.M. contributed equally to the paper. All authors have read and agreed to the published version of the manuscript.

**Funding:** This research received no external funding.

**Institutional Review Board Statement:** Not applicable.

**Informed Consent Statement:** Not applicable.

**Data Availability Statement:** Not applicable.

**Conflicts of Interest:** The authors declare no conflict of interest.

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
