# Peer review of "Optimisation Models for Inventory Management with Limited Number of Stock Items"

_logistics, 2022_

Round 1

Reviewer 1 Report

Dear authors

The subject of inventory management is very important for the whole world, however, we have to delve deeply into the subject or close the scope correctly so as not to be misunderstood, it is necessary to have maturity both on the factory floor, logistics, and data that comes from the market, often companies already work with their own optimization equations, we can't try to simplify everything. I suggest to the authors rethink the writing of the article, to correct the scope that they placed too comprehensive and that perhaps with this large scope there is no optimization solution, because the greater the number of variables, the more complex and time-consuming the optimizations, which may remain in calculations with durations of days, in a question that sometimes the answers have to be in hours or minutes when writing about the topic of optimization, we have to be very careful.

Here are some suggestions:

1) It is necessary to change the title, it is very generic, and you have to close the scope;

2)Improve the abstract;

3)Improve the introduction;

4)Create a new section for a literature review part, and in this section present the state of the art on the subject and have a certain degree of depth in the subject.

5)Improve the example.

6) Improve the Conclusion and create a conclusion with the new text.

Kind regards

Author Response

Point 1: I suggest to the authors rethink the writing of the article, to correct the scope that they placed…. in a question that sometimes the answers have to be in hours or minutes when writing about the topic of optimization, we have to be very careful…. 2) Improve the abstract…

Response 1: The scope of the article is given in the abstract. “The scope of the models in the article is inventory management with a limited number of stock keeping units”.

Point 2: 1) It is necessary to change the title, it is very generic, and you have to close the scope.

Response 2: The title is changed. Old title “An Optimisation Model for Stock Management”. New title: “Optimisation Models for Inventory Management with Limited Number of Stock Items”.

Point 3: 3)Improve the introduction;. …4) have a certain degree of depth in the subject

Response 3: Thanks for the comment. The introduction is extended. “The proposed models by the authors of the article are built with the assumption of limited number of stock items. Companies which have a bug nomenclature of stock items may make a sample of them and analyze it with the proposed models. They need to have observations for real costs to create the historic dataset. This dataset has to be created for each observed item and for fixed time periods, e.g., days, months, quarters.”

Point 4: 4) Improve the example.

Response 4: The model description is precised and extended. “Data for supplied volumes are time series data. Data for the costs of stock management are also time series data. Both time series are with the same time period. Filling in both datasets with this important issue guarantees the correctness of the model. Creating a sample dataset from real business data may be done in 3 steps. Firstly, choosing stock items with fixed-period delivery (e.g., monthly). Secondly, if the list is too big for data collection and data analysis a smaller sample may be chosen. Thirdly, time series data for costs management for each stock unit have to be collected (e.g., monthly). Since companies have confidential data, sample datasets are created by the authors to illustrate the numeric examples”.

Point 5: 6) Improve the Conclusion and create a conclusion with the new text.

Response 5: The conclusion is extended. “The created models have practical implication. Logistics managers may choose a sample from their big items/stocks/inventory list. They may create time series data for delivered volumes (by stocks and by months), time series data for the costs of inventory management (by stocks and by months). They may apply the proposed models to minimize the total costs for stock management. If the optimization models work fast, the logistics managers may extend the sample with more items.”.

Reviewer 2 Report

An interesting attempt to use the methods of mathematical optimization, method of the least squares, regression analysis to determine the most advantageous stock levels for individual types of products in the warehouse. There are doubts, however, which are presented below:

Rows 16-17 Results of research – sentence: „The constructed specific nonlinear mathematical model for optimising the total cost of 16 stock management”. Rows 18-19: „The usefulness of using mathematical models and methods to optimise the total cost of stock management”. Also rows 247-248: „useful tool for determining an optimal strategy for replenishing the stock of multiple products”. The title of article suggests that the research presented concerns optimisation model for overall stock management. Unclear what is final objective of the article and its reflection in the title? This should also be consistently maintained throughout the content of the article.

Rows 59-61: „The aim of the authors in this article is to propose an approach to determine optimal supply sizes from different types of stocks (more than one known in literature as multi-nomenclature) that minimises the overall cost of inventory management”. Does it mean that Authors consider cost of inventory management? Or consideration concern cost of inventory? There are same doubts in further parts of the content.

Rows 100-103: „If the optimal stock volume is to be determined so that the costs are minimal, in this situation the solution is obvious: xopt = xio, where i0 it is determined by the condition cio = min {c1, c2, …, cm}.” Does it mean that optimization of stock volume = minimization of cost of inventory? Perhaps the description should indicate other condition/all conditions, that must be met in order to consider that the stock volume is optimal? Suggested to emphasize all the conditions that the optimal inventory level meets.

The research procedure was carried out correctly. List of sources prepared properly.

Author Response

Point 1: Rows 16-17 Results of research – sentence: „The constructed specific nonlinear mathematical model for optimising the total cost of 16 stock management”.

Response 1: The scope of the research and the methods for data collection are added in the abstract. “The scope of the models in the article is inventory management with a limited number of stock keeping units. Time series data for the delivered quantities and time series data for costs for stock management are used. Both time series are with the same time period”.

Point 2: Rows 18-19: „The usefulness of using mathematical models and methods to optimise the total cost of stock management”…. Also rows 247-248: „useful tool for determining an optimal strategy for replenishing the stock of multiple products”.

Response 2: The sentence in the abstract is revised. “The created mathematical models and methods to optimise the total cost of stock management may be used by logistics managers to minimise the total costs of inventory management”. The conclusion (at the end of the article) is also extended with the practical implication. “The created models have practical implication. Logistics managers may choose a sample from their big items/stocks/inventory list. They may create time series data for delivered volumes (by stocks and by months), time series data for the costs of inventory management (by stocks and by months). They may apply the proposed models to minimize the total costs for stock management. If the optimization models work fast, the logistics managers may extend the sample with more items.”

Point 3: The title of article suggests that the research presented concerns optimisation model for overall stock management. Unclear what is final objective of the article and its reflection in the title? This should also be consistently maintained throughout the content of the article.

Response 3: The title is changed. Old title “An Optimisation Model for Stock Management”. New title: “Optimisation Models for Inventory Management with Limited Number of Stock Items”.

Point 4: Rows 59-61: „The aim of the authors in this article is to propose an approach to determine optimal supply sizes from different types of stocks (more than one known in literature as multi-nomenclature) that minimises the overall cost of inventory management”. Does it mean that Authors consider cost of inventory management? Or consideration concern cost of inventory? There are same doubts in further parts of the content.

Response 4: The abstract is revised. “…that minimises only the cost of inventory management. The cost of inventory is not included.” The same revisions are done in the introduction section, where the aim of the article is presented.

Point 5: Rows 100-103: „If the optimal stock volume is to be determined so that the costs are minimal, in this situation the solution is obvious: xopt = xio, where i0 it is determined by the condition cio = min {c1, c2, …, cm}.” Does it mean that optimization of stock volume = minimization of cost of inventory? Perhaps the description should indicate other condition/all conditions, that must be met in order to consider that the stock volume is optimal? Suggested to emphasize all the conditions that the optimal inventory level meets.

Response 5: The article is revised. The description is given precisely. “The costs are one-dimensional array {c1, c2, …, cm}. The volumes of deliveries are also a one-dimensional array {x1, x2, …, xm}. A possible algorithm finding the optimal solution is the following. The minimal element in the array is found {c1, c2, …, cm}. Its position in the array is stored in the variable “pos”. The optimal volume for stock de-livery is xpos.”.

Reviewer 3 Report

In this paper, the author(s) presented an important logistical problem. The topic is interesting. The presented methodology has great potential in the decision-making process. The solutions to the research problem contained in the text complement the scientific literature in the scope of the present issues. I support the author(s) in researching this topic. However, the author(s) need to consider the following points as limitation or further scope for refining the paper:

1)To add the structure of the paper at the end of the introductory section (The summary of the paper per section - For instance, Section 1 deals with...Section 2 is the review, etc.).

2) Conclusion does not provide any novel findings please elaborate novelty of your work in conclusions. Lines for further research should be identified.

Author Response

Point 1: 1)To add the structure of the paper at the end of the introductory section (The summary of the paper per section - For instance, Section 1 deals with...Section 2 is the review, etc.).

Response 1: The structure of the paper is given. “Section 1 deals with the literature review and the current state of the research problem. Section 2 describes the proposed models, methods and the research design. Results are given in section 3. Conclusions and outline for further research are given in section 4”.

Point 2: 2) Conclusion does not provide any novel findings please elaborate novelty of your work in conclusions

Response 2: The conclusion is extended. “The created models have practical implication. Logistics managers may choose a sample from their big items/stocks/inventory list. They may create time series data for delivered volumes (by stocks and by months), time series data for the costs of inventory management (by stocks and by months). They may apply the proposed models to minimize the total costs for stock management. If the optimization models work fast, the logistics managers may extend the sample with more items.”

Point 3:  Lines for further research should be identified (in the Conclusion).

Response 3: The conclusion is extended. “Further research may be focused on creating new models based on the proposed ones where the time series data for delivery volumes has different time periods than the time series for the costs of inventory management (not including the costs of the inventory itself).”

Reviewer 4 Report

The aim of the authors in this article is to propose an approach to determine optimal supply sizes from different types of stocks

"Methods: The article used the methods of mathematical optimisation, method of the least squares, regression analysis, etc".- Why the authors have written etc. in the abstract. how many different methods have been adopted data need to be furnished. 

The authors have used the sample data. how the sample data is collected/created what is the method of collection of data. - data need to be furnished. 

The research objective needs to be mentioned clearly in the Introduction section. 

What assumptions are made to make the model, data need to be furnished.

What are Sets, decision variables, and parameters, are used need to be mentioned in section 2.

 Research has been done in this area including 10.4018/IJISSCM.2018070104; https://doi.org/10.1108/JMTM-02-2018-0058 which should be updated in the manuscript. 

  In order to elevate the importance and urgency of this study, the author needs to compare their proposed model with others.

Why are there no acknowledgements of previous studies' results? Try adding some validation of your results with reference to the literature such as https://doi.org/10.1007/s10668-021-01713-5.

The research gap needs to be mentioned clearly. 

The novelty of the paper should be highlighted more clearly in all sections. 

The result and conclusion section is very weak. the result needs to be elaborate. 

The conclusion section looks like a summary, there are no in-depth insights in this section, and the managerial implications are also not addressed.

The entire paper has numerous mistakes in sentence structure, grammar, and punctuation (ex: commas, semicolons).  

Author Response

Point 1: "Methods: The article used the methods of mathematical optimisation, method of the least squares, regression analysis, etc".- Why the authors have written etc. in the abstract. how many different methods have been adopted data need to be furnished..

Response 1: We have deleted “etc.”. The scope of the article is precised.

Point 2: The authors have used the sample data. how the sample data is collected/created what is the method of collection of data. - data need to be furnished.

Response 2: The abstract is extended. “Time series data for the delivered quantities and time series data for costs for stock management are used. Both time series are with the same time period”. Detailed information on data collection is given in section 2.2. “Data for supplied volumes are time series data. Data for the costs of stock management are also time series data. Both time series are with the same time period. Filling in both datasets with this important issue guarantees the correctness of the model. Creating a sample dataset from real business data may be done in 3 steps. Firstly, choosing stock items with fixed-period delivery (e.g., monthly). Secondly, if the list is too big for data collection and data analysis a smaller sample may be chosen. Thirdly, time series data for costs management for each stock unit have to be collected (e.g., monthly). Since companies have confidential data, sample datasets are created by the authors to illustrate the numeric examples.”.

Point 3: The research objective needs to be mentioned clearly in the Introduction section.

Response 3: The aim in the introduction section is precised. “The aim of the authors in this article is to propose an approach to determine optimal supply sizes from different types of stocks (more than one known in literature as multi-nomenclature) that minimises only the cost of inventory management. The cost of inventory is not included.”

Point 4: What assumptions are made to make the model, data need to be furnished.

Response 4: The introduction section is extended. “The proposed models by the authors of the article are built with the assumption of limited number of stock items. Companies which have a bug nomenclature of stock items may make a sample of them and analyse it with the proposed models. They need to have observations for real costs to create the historic dataset. This dataset has to be created for each observed item and for fixed time periods, e.g., days, months, quarters.”

Point 5: In order to elevate the importance and urgency of this study, the author needs to compare their proposed model with others.

Response 5: It is done in the literature review section.

Point 6: The novelty of the paper should be highlighted more clearly in all sections.

Response 6: We accept the recommendation. We have thoroughly revised the paper.

Point 7: The result and conclusion section is very weak. the result needs to be elaborate…. The conclusion section looks like a summary, there are no in-depth insights in this section, and the managerial implications are also not addressed.

Response 7: The conclusion is extended. Practical implications are given. Aspects for further research is outlined.

Point 8: The entire paper has numerous mistakes in sentence structure, grammar, and punctuation (ex: commas, semicolons).

Response 8: Thanks for the reviewer. We had a detailed view on the whole paper. A revised version with “track changes” is uploaded to the main editor of the journal.

Round 2

Reviewer 1 Report

Dear Authors

Now the title is good.

The article is improving.

Regarding the rest of the article.

1) The text is better with the Introduction in a separate chapter from the literature review. However, it has to improve both the introduction and the literature review.

2) All Figures 1 to 4 are blurred, the figures from the previous version were much better. It is mandatory to correct.

3) It is necessary to improve and deepen chapter 3 "Results"

Regards

Author Response

Point 1. The text is better with the Introduction in a separate chapter from the literature review. However, it has to improve both the introduction and the literature review.

Response 1. Thank you. We have corrected the article dividing the introduction from literature review, extending and precising the focus.

Point 2. All Figures 1 to 4 are blurred, the figures from the previous version were much better. It is mandatory to correct.

Response 1. Thank you. We will upload DOCX and PDF files.

Point 3.  It is necessary to improve and deepen chapter 3 "Results"

Response 3. We have extended the title of the "Results" section. Now it is "Results and discussion". We have added text, outlining the findings, solving the solutions and marking novelties.

Reviewer 4 Report

1. Figure quality is not good kindly change Fig.

2. Result section needs to be improved not much information is provided by the authors 

3. Authors should compare the result with some benchmark problems to justify the result such as those given in10.4018/IJISSCM.2018070104, and https://doi.org/10.1007/s10668-021-01713-5. 

4. The novelty of the paper should be highlighted more clearly in all sections.

Author Response

Point 1. Figure quality is not good kindly change Fig.

Response 1.Thank you. We will upload DOCX and PDF files. The figures should be OK.

Point 2. Result section needs to be improved not much information is provided by the authors.

Response 2. We have extended the title of the "Results" section. Now it is "Results and discussion". We have added text, outlining the findings, solving the solutions and marking novelties.

Point 3. Authors should compare the result with some benchmark problems to justify the result such as those given in10.4018/IJISSCM.2018070104, and https://doi.org/10.1007/s10668-021-01713-5. 

Response 3. The literature review section is extended focusing on recent models for inventory management.

Point 4. The novelty of the paper should be highlighted more clearly in all sections.

Response 4. Thank you. All sections are extended marking the novelties in the paper.

Round 3

Reviewer 1 Report

Dear Authors

The article got better, it is only necessary to fix the figures from 1 to 4, the figures from version 1 were much better.

Regards